# Common Beverage Consumption and Benign Gynecological Conditions

Rachel Michel [1], Dana Hazimeh [2], Eslam E. Saad [2], Sydney L. Olson [2], Kelsey Musselman [2], Eman Elgindy [3] and Mostafa A. Borahay [2,*]

[1] Department of Population, Family, and Reproductive Health, Bloomberg School of Public Health, Baltimore, MD 21205, USA; rmiche10@jh.edu

[2] Department of Gynecology and Obstetrics, Johns Hopkins University School of Medicine, Baltimore, MD 21205, USA; dhazime1@jhmi.edu (D.H.); esaad2@jh.edu (E.E.S.); solson17@jh.edu (S.L.O.)

[3] Department of Gynecology and Obstetrics, Zagazig University School of Medicine, Zagazig 44519, Egypt

[*] Correspondence: mboraha1@jhmi.edu; Tel.: +1-(410)-550-0337

**Abstract:** The purpose of this article is to review the effects of four commonly consumed beverage types—sugar-sweetened beverages (SSBs), caffeinated beverages, green tea, and alcohol—on five common benign gynecological conditions: uterine fibroids, endometriosis, polycystic ovary syndrome (PCOS), anovulatory infertility, and primary dysmenorrhea (PD). Here we outline a plethora of research, highlighting studies that demonstrate possible associations between beverage intake and increased risk of certain gynecological conditions—such as SSBs and dysmenorrhea—as well as studies that demonstrate a possible protective effect of beverage against risk of gynecological condition—such as green tea and uterine fibroids. This review aims to help inform the diet choices of those with the aforementioned conditions and give those with uteruses autonomy over their lifestyle decisions.

**Keywords:** uterine fibroids (UF); endometriosis; polycystic ovary syndrome (PCOS); anovulatory infertility; primary dysmenorrhea (PD)





## 1. Introduction

Gynecological conditions contribute to approximately 4.5% of the overall global disease burden, exceeding other major global health focuses, including both malaria and tuberculosis [1]. Understanding the impact of nutrition on gynecological conditions is imperative for individuals to make informed diet and lifestyle choices. However, with constantly changing food and drink markets, it can be difficult for consumers to know how certain beverages affect their gynecological health. Thus, the purpose of this review is to investigate four different beverage types—sugar-sweetened beverages (SSBs), caffeinated beverages, green tea, and alcohol—and their effects on five benign gynecological conditions including uterine fibroids (UF), endometriosis, polycystic ovary syndrome (PCOS), anovulatory infertility, and primary dysmenorrhea (PD). Within roughly the last 30 years, the worldwide consumption of SSBs has increased by nearly 16% [2]. While the association between SSBs, obesity, and cardiometabolic diseases has been widely studied, the effect of SSBs on gynecological conditions has not been as thoroughly investigated. Further, caffeine has become the most widely used drug in the world due to its psychoactive effects. With more than 90% of adults consuming it regularly, its high frequency of use has necessitated investigating the unintended effects of caffeine on gynecological health [3]. Also widely consumed, green tea's rich variety of essential nutrients, such as minerals, vitamins, and carotenoids, contribute to its benefits, including for gynecological health [4–7]. Catechins, polyphenolic compounds, in green tea are the focus of different research efforts on the beneficial effects of green tea on health [8,9]. Unlike other types of tea, green tea leaves

are not fermented during the manufacturing process, which preserves the catechins' antioxidant properties [10–12]. Lastly, alcohol use is a common habit worldwide and plays an important role in cultural, social, and celebratory contexts. While alcohol consumption varies regionally, the World Health Organization reports that the global average alcohol consumption rate is roughly 6.2 L of pure alcohol per person 15 years and older, highlighting its prevalent use [13]. Thus, this review aims to increase the transparency of the possible effects of these four beverages on five benign gynecological conditions.

## 2. Uterine Fibroids

Uterine fibroids (UF) are monoclonal neoplasms of the myometrium [14]. They are the most common benign pelvic tumor and can affect up to 80% of reproductive-age individuals with uteruses, with incidence rates highest among African Americans [15,16]. Although many people with fibroids are asymptomatic, around 20–50% may present with symptoms including pelvic pain or pressure, heavy menstrual bleeding, constipation, urinary frequency, and anemia [17]. UF have also been associated with an increased risk of subfertility and pregnancy complications like preterm labor, postpartum hemorrhage, malpresentation, and cesarean section [17–19]. Current treatment modalities range from drug therapies like hormonal contraceptives or gonadotropin-releasing hormone agonists and antagonists, to surgical options like hysterectomy or myomectomy [20]. The following sections will highlight the role of SSBs, caffeinated beverages, green tea, and alcohol on the development of UF.

### 2.1. Uterine Fibroids and Sugar-Sweetened Beverages

The association between SSB consumption and risk of UF is not yet thoroughly investigated. In a prospective study of 21,861 premenopausal individuals, high dietary glycemic index and glycemic load were found to be associated with an increased risk of UF, but the presence of SSBs in participants' diet was unclear [21]. In addition, while this study employed rigorous methods and adjusted for many lifestyle factors in their analysis, dependence on measuring long-term nutrient intake using a yearly food-frequency questionnaire (FFQ) generally biases diet-disease associations.

Excess sugar intake can drive conditions like hyperinsulinemia and insulin resistance, stimulating increased ovarian hormone production, which may indirectly increase development of fibroids [22,23]. In addition, sugar intake from SSBs may cause inflammation and angiogenesis, all of which promote a pro-tumorigenic environment (Figure 1) [24–31]. Further research is necessary to fully illuminate the effects of overconsumption of SSBs on UF development.

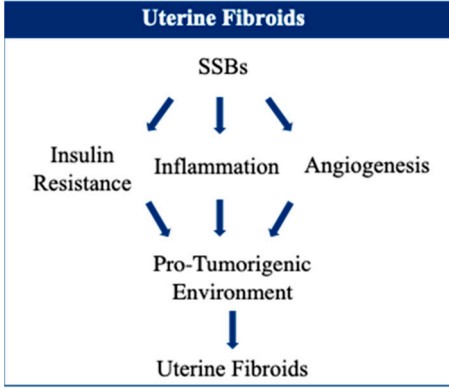

**Figure 1.** Effect of sugar-sweetened beverage (SSB) consumption on pro-tumorigenic pathways contributing to fibroid development.

### 2.2. Uterine Fibroids and Caffeinated Beverages

Due to its high consumption, multiple studies have investigated the role of caffeine and risk of UF. The Black Women's Health Study showed an increased risk of UF only

among women aged <35 years who consume at least 500 mg of caffeine per day, however since caffeine metabolism does not differ by age in humans, the authors suggest caution in interpreting this result [32]. The Seveso Women's Health Study found a higher prevalence of UF among those who drank at least one cup of coffee per day, however this analysis was not adjusted for other covariates [33]. An earlier Italian study did not find an association between caffeine intake and risk of UF development, however due to its case-control design, recall bias may be present in self-reporting of diet [34].

Some studies suggest that caffeine consumption impacts sex-hormone levels which may in turn impact hormone-driven tumors like leiomyoma. In a cross-sectional study, Lucero et al. measured early follicular phase hormones in 498 individuals aged 36–45. The study found a significant association between total caffeine use and estradiol levels, even when adjusting for age, BMI, total caloric intake, current smoking status, alcohol, cholesterol consumption, and day of sampling. Moreover, they observed a 70% increase in early follicular phase estradiol levels with high caffeine consumption (500 mg per day or more) compared to milder consumption (100 mg per day or less) [35]. This may be mediated by the interaction of caffeine with enzymes that are responsible for estrogen metabolism [36,37]. Specifically, caffeine is thought to exhibit an inhibitory effect on the phosphodiesterase enzyme resulting in increased levels of cyclic adenosine monophosphates which enhances sex steroids production [38]. A study in 2015 explored the association between caffeine consumption and mid-luteal phase urine levels of 15 individual estrogens and estrogen metabolites among premenopausal individuals. Their findings corroborate existing literature, suggesting that caffeine plays an important role in modulating the metabolism and levels of estrogen [39]. Caffeine's biological role and relation to estrogen provides insight into its possible effect on uterine leiomyoma, but more clinical studies are needed to draw definitive conclusions.

### 2.3. Uterine Fibroids and Green Tea

Epigallocatechin gallate (EGCG) is a type of green tea catechin that has been extensively studied in both benign and malignant gynecologic disorders [40,41]. EGCG has been shown to promote apoptosis and inhibit the proliferation of leiomyoma cells in vitro [42–44]. When administered to animal models including female athymic nude mice and Japanese quail, EGCG was found to significantly reduce the size of UF [42,45]. EGCG also has promising results in human studies. In a randomized double-blinded clinical trial of 39 individuals with symptomatic UF, participants were either given 800 mg of green tea extract or placebo daily for 4 months. The authors reported a 32.6% reduction in fibroid volume as well as significant improvement in symptom profile and quality of life [46]. Recent Italian pilot studies have also supported these findings, where administration of EGCG, in combination with other vitamins, also resulted in a significant reduction in fibroid tumor size [47,48].

The mechanism of action of EGCG is not yet fully understood. One theory focuses on EGCG's inhibitory action on the Catechol-o-methyltransferase (COMT) enzyme. The COMT enzyme is upregulated in leiomyoma cells and increases estrogen levels contributing to fibroid development [49,50]. In addition to targeting COMT, EGCC is also thought to modulate bone morphogenetic protein-2 (BMP2) expression, which has been associated with fibroid size. By increasing BMP2 levels and consequently reducing transforming growth factor beta (TGF-β) levels, EGCG has been postulated to control fibroid growth [51]. A recently published study on human fibroid cells treated with EGCG noted its effects on several key signaling pathways involved in fibroid pathogenesis. EGCG treatment reduced the levels of cyclin D1, a protein that is increased in leiomyoma cells and promotes cell cycle progression. Moreover, EGCG treatment exerted anti-fibrotic effects on leiomyoma cells by altering the activation of pro-fibrotic signaling pathways (yes-associated protein 1 (YAP), β-catenin, c-Jun N-Terminal Kinase (JNK) and protein kinase B (AKT) pathways), and significantly decreasing the expression of key fibrotic proteins like fibronectin (FN1), collagen (COL1A1), plasminogen activator inhibitor-1 (PAI-1), connective tissue growth

factor (CTGF), and alpha-smooth muscle actin (ACTA2) in fibroid cells [52]. Ongoing clinical trials are examining the efficacy and safety of ECGC found in green tea (NCT04177693; NCT05409872; NCT0536008) and are further illuminating of its promising role in UF treatment and prevention.

### 2.4. Uterine Fibroids and Alcohol

The available data begin to shed light on the relation between alcohol consumption and UF risk, however many limitations inhibit drawing definite conclusions. The Black Women's Health Study found both current heavy alcohol consumption status and years of alcohol consumption to be positively associated with an increased risk of UF development, with incidence rate ratios (IRRs) of 1.57 (95% CI: 1.17, 2.11) for >7 glasses, 1.18 (95% CI: 1.00, 1.40) for 1 to 6 glasses, and 1.11 (95% CI: 0.98, 1.27) for <1 glass of alcohol per week. Interestingly, this association was stronger with beer consumption compared to wine or liquor consumption, possibly due to the presence of phytoestrogens in beer and its differential effect on hormone-dependent tumors like UF [32]. Consistent findings were reported in the California Teachers Study (CTS) where participants who consumed 20 g of alcohol or more daily had a significantly increased risk of surgical treatment for UF (risk ratio (RR): 1.33; 95% CI: 1.12, 1.58) [53]. A cross-sectional study in Japan also reported a significant positive association between alcohol intake and leiomyoma prevalence among 285 premenopausal individuals coming in for a routine health-checkup [54]. Similarly, the Seveso Women's Health study in Italy found that the prevalence of UF was significantly higher among those with a history of previous or current alcohol consumption [33]. More recently, a retrospective nationwide population-based cohort study in Korea demonstrated a positive association between alcohol consumption and UF. The authors found that alcohol consumption was associated with an increased incidence of new-onset UF with a statistically significant hazard ratio (HR) ranging from 1.12 (95% CI: 1.11, 1.14) for mild-to-moderate drinkers to 1.16 (95% CI: 1.12, 1.20) for heavy drinkers. Moreover, the authors reported a dose-dependent relationship whereby the incidence of new-onset UF increased proportionately to the number of alcohol glasses consumed per drinking session [55]. However, due to the exploratory design and reliance on self-reported alcohol intake across these studies, the evidence is not robust enough to suggest a causal effect.

The exact etiopathogenesis behind the role of alcohol in UF development is not yet fully understood. One plausible mechanism is alcohol's effect on estrogen hormone levels, which is known to promote UF development and growth [56–59]. UF tissue is characterized by higher concentrations of estrogen, as well as a higher expression of aromatase and estrogen receptor compared to the normal myometrium [60,61]. Alcohol, in turn, can promote the hormone-dependent growth of leiomyoma tissue through several possible pathways. First, alcohol can reduce estrogen metabolism thereby increasing the levels of endogenous estrogen; second, alcohol can interact with luteinizing hormone (LH) to modulate estrogen release from the ovaries; and third, alcohol is thought to induce the activation of aromatase enzyme as well as estrogen receptor; all of which may play a role in UF development [56,62–67]. Overall, the aforementioned findings support the potential for a positive association between uterine fibroids and alcohol consumption, but current data are limited in their design to fully draw this connection.

## 3. Endometriosis

Endometriosis is a benign chronic gynecologic disorder affecting about 6–10% of reproductive aged patients and is characterized by implantation of endometrial-like lesions outside of its normal location [68–71]. It is commonly identified in those with chronic pelvic pain, dysmenorrhea, and/or infertility, and can significantly affect quality of life [72]. Endometriosis is thought to arise from retrograde menstruation or cellular metaplasia; however, much is still not understood regarding its etiology or underlying mechanisms of its spread, symptoms, or impact on fertility. There is no racial predisposition for developing endometriosis, however a familial link does appear to be at play, as individuals with an

affected first-degree relative are 7–10 times more likely than the general population to develop endometriosis [70].

Endometriosis is diagnosed via surgical pathology, and once found should be treated as a chronic condition with a high recurrence rate even after surgical resection [69,72]. Treatment options for pain associated with endometriosis include non-steroidal anti-inflammatory drugs (NSAIDs), combined estrogen-progesterone oral contraceptives, oral or injectable progestins, levonorgestrel intrauterine devices, gonadotropin-releasing hormone agonists and antagonists, and surgery [68,72,73]. More recently, endometriosis is being considered a systemic disease beyond the pelvis which may affect metabolism, inflammation, and alter gene expression [25,31,71,74]. As such, diet and nutritional factors may play a significant role in pathophysiology and symptomatology of endometriosis [75]. The following sections will review available data on effect of sugar-sweetened beverages, green tea, alcohol, and caffeine on endometriosis.

### 3.1. Endometriosis and Sugar-Sweetened Beverages

Currently, we are not aware of any studies investigating the role of SSB consumption and endometriosis. Due to the proinflammatory properties of sugar, it is reasonable to expect some sort of interplay between sugar consumption and risk of endometriosis [76]. Some studies document the importance of a healthy, balanced diet for those with endometriosis, however there is little to no emphasis on sugar intake specifically [77,78]. One study investigating the influence of metabolic indexes on the number of retrieved oocytes and assisted reproductive technology (ART) outcomes in patients with endometriosis found that serum insulin levels were related to the number of retrieved oocytes, and both serum glucose and insulin levels were related to the occurrence of gestational diabetes [79]. This study highlights the role of dysregulated glucose metabolism on fertility outcomes, however further study exploring the effects of SSB consumption on risk of endometriosis would be of great value.

### 3.2. Endometriosis and Caffeinated Beverages

While found in numerous foods and beverages, the data on caffeine and its association with endometriosis are mixed. Caffeine consumption has been associated with higher sex hormone-binding globulin concentrations and lower testosterone levels and has been thought to potentially negatively impact hormone-dependent diseases such as endometriosis [80]. Looking at population data, one large cross-sectional study of patients with endometriosis showed an association between coffee consumption and endometriosis in crude analysis but not the adjusted analysis [81]. In addition, multiple studies have not found a correlation between surgically confirmed endometriosis and caffeine or coffee consumption [82–84]. One meta-analysis examining relative risk of endometriosis to any, high, and low versus any coffee/caffeine consumption did not demonstrate an association between caffeine and the risk of endometriosis [85]. Conversely, a more recent meta-analysis showed that high caffeine consumption (over 300 mg/day) significantly increased the risk of having endometriosis compared to little (less than 100 mg/day) or no caffeine (RR 1.30, 95% CI: 1.04, 1.63) [86]. Moderate caffeine (100–300 mg/day) and overall caffeine intake compared to little or no caffeine also showed modest increases in risk of endometriosis but the difference was not statistically significant [86]. Overall, due to the inconsistent data across findings and lack of clinical data to draw causal conclusions, future animal and human studies are necessary to fully elucidate the role of caffeinated beverage consumption and risk of endometriosis.

### 3.3. Endometriosis and Green Tea

In addition to its positive effects on UF development, EGCG, the major catechin in green tea, is thought to potentially alleviate severity of endometriosis-related symptoms through its anti-angiogenic, pro-apoptotic, and anti-fibrotic properties [7,41,87]. Multiple studies utilizing in vivo mouse models with transplanted mouse or human endometrium

to other anatomical locations have demonstrated regression of endometriosis cells after initiation of EGCG treatment compared to placebo [88–93]. The results from these studies reliably showed that EGCG was associated with decreased growth of endometrial implants, smaller lesion size and weight, and prevention of new lesion development [88–90,92,93].

EGCG may exhibit antiangiogenic effects on endometriosis by impeding vascular epithelial growth factor (VEGF) signaling. EGCG in vitro and in vivo mouse models have been associated with downregulated expression of pro-angiogenic cellular signaling pathways leading to lower microvessel size and density [88,89,91] (Figure 2). Pro-angiogenic factors like VEGF-A, VEGF-C, and tyrosine kinase receptor VEGF receptor 1 (VEGFR2) had lower expression after treatment with EGCG [89,91].

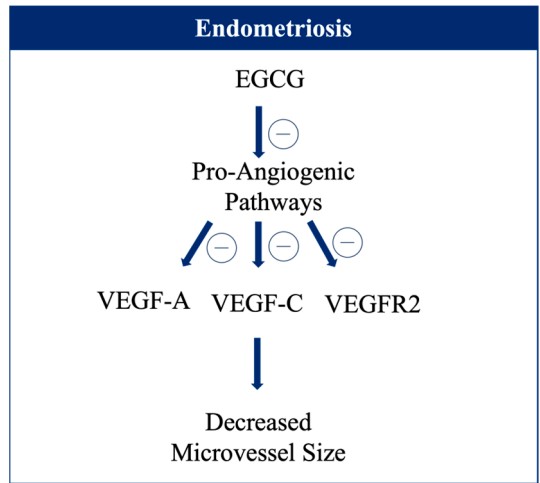

**Figure 2.** Effect of Epigallocatechin gallate (EGCG) on pro-angiogenic cellular signaling pathways, which acts by downregulating vascular epithelial growth factors (VEGF).

Research on the effects of green tea on endometriosis in humans remains limited. A phase 2 randomized controlled trial was recently conducted in Hong Kong where patients were orally administered either 400 mg of EGCG or placebo twice per day to see whether there was a change in endometriotic lesion size or improvements in pain and quality of life, however this clinical trial has yet to publish any results (NCT02832271). Overall, due to the promising results from animal models of EGCG treatment, green tea may potentially alleviate endometriosis related symptoms, but results from clinical trials are necessary to fully support this conclusion.

### 3.4. Endometriosis and Alcohol

As stress and inflammation pathways likely play an important role in endometriosis onset and symptoms, alcohol has been explored as a possible modifiable risk factor for endometriosis. Alcohol intake is thought to alter reproductive hormones through aromatase activation, which promotes conversion of testosterone to estrogen in peripheral adipose tissue [82]. Much of the data regarding dietary alcohol consumption and risk of endometriosis are mixed. One retrospective case-control study comparing patients with endometriosis to infertile patients without endometriosis on laparoscopy demonstrated an increased incidence of endometriosis in those who consumed alcohol [94]. However, several other case-control and cohort studies have concluded that there may not be a significant association [82,83]. Interestingly, one prospective cohort study of over 1700 individuals and a smaller observational study showed an inverse relationship between the amount of alcohol intake and endometriosis [84,95] Further, two meta-analyses performed by the same group looked at alcohol intake and endometriosis based on the literature available in 2012 and 2021 [96,97]. Their findings initially suggested an association between endometriosis and alcohol with a summary estimate OR of 1.24 (95% CI: 1.12, 1.36) [96]. However, the updated meta-analysis from 2021 instead demonstrated only borderline statistical significance when

comparing any alcohol consumption to none (OR 1.14; 95% CI: 0.99, 1.31, *p* = 0.06), though it did find a significant association with moderate alcohol intake of less than 7 drinks per week and endometriosis (OR 1.22, 95% CI: 1.03, 1.45, *p* = 0.02) [97]. Interpretation of these case-control and cohort studies should be done with caution due to limitations in their design. Studies using animal models would be of value to draw potential conclusions regarding the relationship between alcohol consumption and risk of endometriosis.

## 4. Polycystic Ovary Syndrome

Polycystic ovary syndrome (PCOS) is a condition characterized by hyperandrogenism, ovulatory dysfunction, and polycystic ovaries affecting up to 6–20% of reproductive aged individuals with uteruses worldwide [98–100]. PCOS is a leading cause of female factor infertility, contributing to estimated costs of $8 billion annually (2020) in diagnosis, treatment, and infertility services the United States [101,102]. The etiology of PCOS is multifactorial and is thought to involve genetic traits which either manifest as clinically evident disease or establish a pre-disposition that may progress to PCOS when exposed to environmental triggers. Inherited genetic traits lead to elevated levels and action of androgens which interfere with normal ovulatory cycles leading to multiple immature ovarian cysts, giving the disease its name (Figure 3) [103,104]. Hyperinsulinemia and insulin resistance are other key features of PCOS, as well as increased rates of metabolic syndrome, gestational diabetes mellitus, impaired glucose tolerance (IGT) and T2DM [105,106]. Thus, diet choices and development of obesity are thought to trigger onset of PCOS in genetically susceptible individuals [107]. No specific recommendations have been made on beverage choices in the most recent international evidence-based guidelines of managing PCOS; the following sections will review available data on effect of SSBs, green tea, alcohol, and caffeine on PCOS [108].

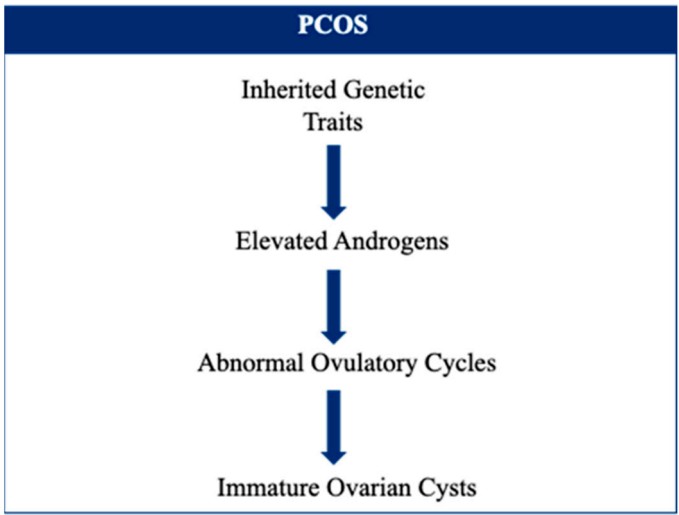

**Figure 3.** Mechanism of polycystic ovary syndrome (PCOS) onset.

### 4.1. PCOS and Sugar-Sweetened Beverages

The association between obesity and PCOS continues to be studied both for the role of obesity inducing PCOS in susceptible individuals as well as PCOS metabolic changes contributing to obesity. Numerous studies have supported healthy diet interventions in improving insulin resistance and metabolic markers in PCOS, but few have specifically studied the effect of SSBs [109]. One study in a PCOS rodent model demonstrated that high-sugar, high-fat diet induces polycystic ovaries, irregular estrous cycles, and hyperinsulinemia: supporting diet as an inciting factor in development of PCOS [110]. Another study with a similar rodent model corroborated these findings, also concluding that a high-sugar, high-fat diet impairs basal hormone levels, which was strongly correlated with ovarian cyst formation [111]. Similarly, an ecologic study of state level data in Brazil using

lag analysis found that consumption of SSBs was associated with increased prevalence of PCOS 10 years later [112]. Further, a recently published review article detailed how oxidative stress stimulated by sugar consumption may be a driver of hyperandrogenism in PCOS patients [113]. Overall, an association between SSB consumption and PCOS is supported by these data, but further animal and clinical studies are required to draw a definite causal relationship.

*4.2. PCOS and Caffeinated Beverages*

Limited studies have looked at the effect of caffeine consumption on PCOS specific patient populations. Caffeine has been shown to increase cortisol, which increases insulin resistance and decreases progesterone and would theoretically worsen symptoms of PCOS, but this hypothesis has not been rigorously studied [114]. One study of a rat model for PCOS found that caffeine considerably reduced ovary volume and follicular clusters but increased inflammatory markers interleukin 6 (IL-6) and tumor necrosis factor $\alpha$ (TNF$\alpha$) [115]. Interestingly, one case-control study found that coffee consumption was associated with reduced risk of PCOS ($p < 0.05$), however due to the limited nature of the design, strong conclusions cannot be made [116]. Further studies are needed on PCOS specific populations to fully understand caffeine's role in disease progression.

*4.3. PCOS and Green Tea*

The role of green tea on PCOS onset and display of symptoms has yet to be fully elucidated. To date, 4 high quality double-blinded placebo control trials for green tea have been conducted in patients with PCOS. Mombaini et al. studied 500-mg green tea (*C. Sinensis* L.) leaf powder ($n = 22$) versus corn-starch tab ($n = 23$) taken for 45 days and found a significant reduction in BMI, waist circumference, and percent body fat in the green tea group; although the reduction was not significantly different compared to the placebo group. This study found no significant reduction in serum inflammatory marks IL-6, C-reactive protein, and TNF$\alpha$ [117]. Tehrani et al. analyzed 1000 mg green tea experimental (GTE) group for 12 weeks and found significant reduction in weight, mean fasting insulin, and free testosterone in the GTE group ($n = 30$) compared to placebo ($n = 30$) [118]. A third study by Farhadian et al. of 1500 mg GTE for 12 weeks found significant reductions in BMI and waist-circumference over the study, and no difference in placebo and metformin 1500 mg daily groups [119]. In contrast, a study on green tea supplementation in obese Chinese women with PCOS did not find a significant reduction of body weight compared to controls, nor did it find an altered biochemical profile [120]. A systematic review meta-analysis of these four trials identified that green tea group had significantly lower body weight at the end of the trials compared to placebo [121]. Overall, green tea demonstrates potential as a therapy for patients with PCOS, but more rigorous studies are needed to further support this.

*4.4. PCOS and Alcohol*

The role of alcohol as a contributor to PCOS development is not clear. The Australian Longitudinal Study on Women's Health (ALSWH) did not identify different rates of alcohol use between those diagnosed with PCOS and those without, suggesting a limited role of alcohol as a trigger for developing PCOS [122]. Interestingly, a study aimed at synthesizing evidence of dietary behaviors between those with PCOS and those without found that individuals with PCOS had lower alcohol consumption compared to controls (CI: $-1.67$, $-0.22$) [123]. Overall, these studies are limited in their design and more rigorous methods are needed to illuminate the relationship between PCOS and alcohol.

**5. Anovulatory Infertility**

Anovulation is a disorder where the intricate process of releasing the ovum from the follicle is disrupted, contributing to infertility [124]. Normally, mammalian reproductive functions are regulated by the hormonal hypothalamic-pituitary-ovarian axis (HPO), which

involves the interaction of hypothalamic gonadotropin-releasing hormone (GnRH), the two pituitary gonadotropins—follicle stimulating hormone (FSH) and luteinizing hormone (LH), and ovary derived hormones (Figure 4). Thus, imbalances in this intricate system have major effects on reproduction, as HPO dysfunction is the leading cause of anovulation (85% of cases) [125]. Diet is thought to play a role in the development of anovulatory infertility, thus the following sections will review available data on effect of SSBs, green tea, alcohol, and caffeine on disease pathology [126].

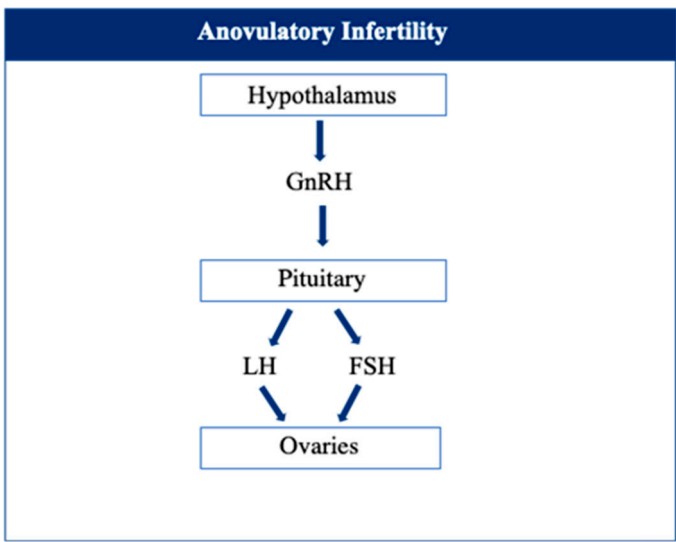

**Figure 4.** Normal hypothalamic-pituitary-ovarian (HPO) axis mechanism involving interaction of hypothalamic gonadotropin-releasing hormone (GnRH), follicle stimulating hormone (FSH) and luteinizing hormone (LH).

*5.1. Anovulatory Infertility and Sugar-Sweetened Beverages*

An association between SSB consumption and reduced fertility has been found across multiple studies. In one study, those who consumed ≥7 SSBs per week were found to have reduced fecundability compared to those who did not (CI: 0.70, 0.94) [127]. Another prospective study investigating dietary factors and fertility outcomes found that total carbohydrate intake and dietary glycemic load were positively associated with ovulatory infertility, even after adjustment for confounders (RR = 1.91, CI: 1.27, 3.02) [128]. Notably, the design of these prospective studies and their reliance on FFQs for reported beverage intake limits the strength of conclusions drawn. Another study found that after ovarian stimulation, a higher intake of SSBs was linked to fewer mature, fertilized oocytes and high-quality embryos [129]. Compared to individuals who did not drink SSBs, those who did had, on average, 1.1 fewer retrieved oocytes, 1.2 fewer mature oocytes retrieved, 0.6 fewer fertilized oocytes, and 0.6 fewer top-quality embryos. Overall, consumptions of SSBs may have a deleterious effect on fertility, but the current research is limited.

*5.2. Anovulatory Infertility and Caffeinated Beverages*

To date, the current data has revealed little to no association between caffeine intake and anovulatory infertility. A prospective cohort study in Denmark assessing beverage intake and time to pregnancy found no significant correlation between fecundability and coffee consumption of three or more servings per day or 300 mg or more of caffeine per day as opposed to less than 100 mg (FR = 1.04 [95% CI: 0.90, 1.21]) [130]. The pooled results from a systematic review and dose-response meta-analysis also showed that there was no correlation between coffee/caffeine consumption and the results of fertility treatment [131]. Chavarro et al. found that caffeine use had no effect on ovulatory dysfunction infertility, with the multivariate-adjusted RR and 95% CI comparing the highest to lowest caffeine intake reported as 0.86 (0.61–1.20), respectively [132]. Lastly, a systematic review from

controlled clinical studies showed that caffeine consumption in low (OR 0.95, 95% CI: 0.78, 1.16), medium (OR 1.14, 95% CI: 0.69, 1.86), and high (OR 1.86, 95% CI: 0.28, 12.22) doses does not increase the risk of infertility [133]. While cohort and case-control studies cannot support causal inference, taken together with clinical data, the role of caffeine intake in risk of anovulatory dysfunction ultimately remains dubious.

### 5.3. Anovulatory Infertility and Green Tea

Because of its potent antioxidant properties, epigallocatechin-3-gallate (EGCG) is regarded as a promising bioactive component in green tea. Multiple review studies have postulated EGCG's potential to improve fertility outcomes by reducing oxidative stress, promoting ovulation, and inhibiting the development of cysts [7,134]. Another study suggested that due to the capacity of catechin polyphenols to squelch reactive oxygen species, green tea supplementation may have the potential to enhance the quality of gametes [135]. Overall, clinical intervention studies are needed to provide clear evidence of role of green tea in improving fertility outcomes, as review studies are not robust enough to draw definite conclusions.

### 5.4. Anovulatory Infertility and Alcohol

Multiple studies have demonstrated an association between alcohol intake and the risk of infertility. In a prospective cohort study of fertility, investigators found that in the luteal phase, heavy intake (>6 drinks per week) and moderate intake (3–6 drinks per week) of alcoholic beverages was respectively associated with a reduced odds of conception and reduced fecundity [136]. In addition, a population-based cohort study of Danish women found that while baseline alcohol use was not linked to infertility in younger individuals, it was a strong predictor of infertility in those over 30 [137]. For this age group, the adjusted hazard ratio for consuming seven or more drinks per week was 2.26 (CI: 1.19, 4.32) compared to those who consumed less than one drink per week, indicating that alcohol intake may be a predictor for infertility among those attempting pregnancy later in life [137]. A third prospective study in Stockholm County, Sweden found that high alcohol consumption was associated with increased risk of infertility examinations at hospitals (RR = 1.59, CI: 1.09, 2.31) [138]. Overall, while an association is notable across multiple cohort studies, without clinical evidence, a definite conclusion on the role of alcohol's contribution to infertility cannot be made.

## 6. Primary Dysmenorrhea

Dysmenorrhea is a common disorder characterized by painful abdominal cramps right before or during the onset of menses [139]. This condition can be classified into primary or secondary dysmenorrhea. Primary dysmenorrhea (PD) is characterized by the experience of painful menstruation, without evidence of pelvic pathology, while secondary dysmenorrhea (SD) refers to painful menstruation resulting from a pelvic or gynecological disorder such as endometriosis or UF [140]. Pain from dysmenorrhea is difficult to measure as there is no instrument or clinical evaluation to confirm incidence, however reported prevalence rates of PD vary between 16–90% of menstruating individuals, making PD the leading cause of recurrent short-term school or work absenteeism [141–144].

The pain associated with dysmenorrhea is attributed to increased secretion of prostaglandins in the endometrium [145]. Specifically, hypersecretion of prostaglandins F2$\alpha$ and E2 have been implicated in the pelvic pain associated with the disorder [139]. In prostaglandin biosynthesis, prostaglandin is produced from arachidonic acid via the cyclooxygenase (COX) pathway [146]. Phase-specific expression of prostaglandins results from COX pathway activation via the progesterone drop during menstruation [147]. Due to prostaglandins' hormone-like properties as a lipid, they are typically responsible for responding to areas of infection or damage. Thus, an overproduction can alter the uterine environment, leading to uterine muscle ischemia, increased nerve sensitivity, and ultimately uterine hypercontractility, resulting in pelvic pain (Figure 5) [148].

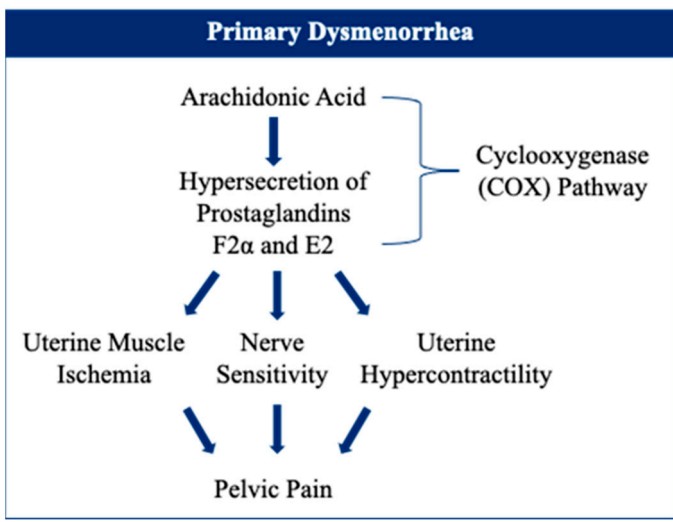

**Figure 5.** Mechanism of primary dysmenorrhea.

In addition to pelvic pain, PD associated symptoms include headache, lethargy, sleep disturbances, tender breasts, back aches, nausea and vomiting, constipation or diarrhea, and increased urination frequency [139]. Treatment options for PD include non-steroidal anti-inflammatory drugs (NSAIDS) alone or in combination with oral contraceptives or progestins, or one may utilize non-pharmacological aids such as heating pads or exercise [139,145]. The following sections will review available data on effect of SSBs, green tea, alcohol, and caffeine on primary dysmenorrhea.

### 6.1. Primary Dysmenorrhea and Sugar-Sweetened Beverages

Refined sugars are considered pro-inflammatory, and it is thought they contribute to prostaglandin synthesis, worsening symptoms associated with dysmenorrhea [149]. Multiple studies have demonstrated an association between SSBs and dysmenorrhea. One retrospective case-control study investigating prevalence and risk factors dysmenorrhea in adolescents in Tbilisi, Georgia found a marked increase of dysmenorrhea in those with an increased daily intake of sugar compared to those reporting no daily sugar intake (55.61% vs. 44.39%, $p = 0.0023$) [150]. Another study investigating the prevalence and predictors of dysmenorrhea among students at a university in Turkey found a 1.8 times higher risk of dysmenorrhea in those with an excessive daily sugar intake compared to those without (OR 1.77; 95% CI: 1.15, 2.72) [151]. A third study investigating the same parameters in preparatory school students in Debre Markos, North-West Ethiopia found that sugar intake was again statistically significantly associated with dysmenorrhea, adjusted OR = 2.94 (CI: 1.54, 5.61) [152]. In addition, a nested case control study investigating the relation between dietary patterns and the risk of dysmenorrhea among students at Urmia University of Medical Sciences found that high consumption of sugars was associated with an increased risk of dysmenorrhea [153]. Although not statistically significant, a study investigating the correlation between diet and dysmenorrhea among high school and college students in Saint Vincent and Grenadines reported dysmenorrhea in a large proportion of participants who consumed high quantities of sugars [154]. Inconsistently however, separate studies have reported no association between intake of SSBs and dysmenorrhea [155,156]. Ultimately, the compiling evidence may point towards reducing SSB intake to reduce symptoms of dysmenorrhea, however the lack of clinical data and limited nature of exploratory studies prevents definite conclusion.

### 6.2. Primary Dysmenorrhea and Caffeinated Beverages

The relationship between caffeinated beverage intake and risk of PD has been supported across multiple cohort and cross-sectional studies. One study examining the prevalence and correlation between diet and dysmenorrhea among high school and college

students in Saint Vincent and Grenadines found a statistically significant correlation between consumption of caffeinated drinks and dysmenorrhea ($p < 0.05$) [154]. This coincides with the results from a cross-sectional study examining the prevalence and predictors of dysmenorrhea among students at a university in Turkey. This study found that the prevalence of dysmenorrhea was significantly higher among those who reported frequent intake of caffeine compared with those who reported infrequent intake (61.0% vs. 48.3%; $p < 0.001$) [151]. A separate study looking at risk factors for primary dysmenorrhea using a sample from university students in Corum, Turkey again found a statistically significant relationship between consumption of caffeinated beverages and the prevalence of dysmenorrhea ($p < 0.05$). They also found the prevalence of dysmenorrhea (86.8%) to be higher in those who regularly consumed high amounts of caffeinated beverages compared to those who did not consume these drinks (13.2%) [157]. While another study investigating the relationship between menstrual pain and the intake of nutrients did not find a significant association between coffee consumption and menstrual pain ($p = 0.809$), this may be due to the study's inclusion of coffee only versus caffeinated beverages as a whole [158]. Notably, these studies are limited in their design and reliance on self-reported caffeine intake. The mechanistic link between caffeine and dysmenorrhea is unclear, however, the potentiation of muscle contraction via induction of sarcoplasmic reticulum calcium release may play a role in the uterine hypercontractility [159]. Overall, more investigation is needed into studying the mechanistic role caffeine may play on the pelvic pain experienced by patients to draw solid conclusions.

### 6.3. Primary Dysmenorrhea and Green Tea

Green tea has garnered attention in recent years for its anti-inflammatory, antioxidative, cardio and neuroprotective properties as well as its cholesterol lowering effects [160]. However, green tea's role in reducing symptoms of dysmenorrhea is much less known [161]. Studies have noted the ability of EGCG to limit prostaglandin biosynthesis by targeting the COX pathway, reducing uterine contractility and overall lessoning the pelvic pain associated with dysmenorrhea [41]. In a cross-sectional study using the Shanghai Birth Cohort, tea drinking was associated with a lower prevalence of dysmenorrhea (adjusted OR [aOR] = 0.68, 95% CI: (0.50, 0.93)) for those with mild dysmenorrhea, with green tea exhibiting the strongest reduction when compared to other tea varieties [162]. A literature review investigating the benefits of green tea consumption on reproductive disorders also cited EGCG's role in reducing hypercontractility of the uterus, improving symptoms of dysmenorrhea [7]. Overall, the possibilities of green tea as a therapeutic option for treatment of pelvic pain is extremely underexplored. While review studies have noted its potential, more clinical trials and animal studies are needed to fully elucidate the benefits green tea may pose on treating patients with dysmenorrhea.

### 6.4. Primary Dysmenorrhea and Alcohol

Despite alcohol's pro-inflammatory properties, the evidence measuring an association between alcohol consumption and dysmenorrhea is not strong [163]. A retrospective case-control study investigating prevalence and risk factors for dysmenorrhea in adolescents in Tbilisi, Georgia found no association between alcohol use and risk of dysmenorrhea [150]. Similarly, a comprehensive systematic review performed on longitudinal, case-control or cross-sectional studies did not find a relationship between alcohol use and risk of dysmenorrhea, supporting previous literature [143,164]. One case-control study in Milan, Italy even found a protective effect of alcohol consumption, and that in comparison with teetotalers, the age-adjusted RR of dysmenorrhea was 0.8 (95% CI: 0.4, 1.5) for alcohol drinkers [165]. However, the authors report that these findings were not estimated with great precision. Oppositely, while a cross-sectional observational study conducted among university students in North China did not find a crude association between alcohol consumption and dysmenorrhea, after stratifying for age at menarche, the authors found a positive association between alcohol consumption and dysmenorrhea among participants

with age at menarche $\geq$13 years (OR 1.41; 95% CI: 1.06, 1.88). In addition, after adjusting for confounders, the authors found that participants who reported consuming alcohol at least once a month (OR 1.29; 95% CI: 0.94, 1.78) and at least once a week (OR 1.92; 95% CI: 1.07, 3.45) were more likely to have dysmenorrhea compared with participants who reported no alcohol consumption [166]. However, the ability to draw causal conclusions based on these investigations is inhibited by their case-control and cohort study design. Notably, while several studies did not find an association between alcohol intake and dysmenorrhea, medical institutions like Johns Hopkins Hospital, Boston's Children Hospital, and Cedars Sinai report alcohol consumption as a risk factor for dysmenorrhea on their websites, suggesting a reduction in alcohol consumption may improve symptoms associated with dysmenorrhea [167–169].

## 7. Limitations

There are notable limitations in this review's ability to draw definite links between beverage intake and gynecological condition. Firstly, this review presents many case-control studies and cohort studies which are limited in their ability to make causal inferences. Next, many case-control and cohort studies discussed rely on FFQs or self-reports of beverages, possibly resulting in misestimation of dietary intake and biasing results of these studies. In addition, studies of these nature cannot always adjust for covariates such as use of hormonal contraception, parity, or other lifestyle factors which may play a role in development of gynecological conditions.

## 8. Conclusions

Overall, there is evidence to suggest associations between beverage intake and gynecological conditions, some of which is notably limited. More animal models and clinical studies are needed to draw casual conclusions. Those predisposed to or with benign gynecological conditions should be informed of the diet factors that potentially influence the aforementioned disorders.

**Author Contributions:** Conceptualization, M.A.B.; investigation, R.M. and M.A.B.; writing—original draft preparation, R.M., D.H., E.E.S., S.L.O. and K.M.; writing—review and editing, R.M., E.E. and M.A.B.; supervision, E.E. and M.A.B. All authors have read and agreed to the published version of the manuscript.

**Funding:** This work was supported, in part, by National Institutes of Health grants R01HD094380 and R01HD111243.

**Acknowledgments:** We gratefully acknowledge fruitful discussions with Mara Michel and Eli Zoghlin.

**Conflicts of Interest:** The authors declare no conflicts of interest.

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
