# Peer review of "Common Beverage Consumption and Benign Gynecological Conditions"

_beverages, doi:10.3390/beverages10020033_

Round 1
Reviewer 1 Report
Comments and Suggestions for Authors
The Review is interesting as it considers the halthy effects of the most four consumed beverages in benign gynecological conditions.
It well readable and well constructed.
I suggest to avoid the addition of a Table in Conclusion section. It could be better to add it at the end of the discussion section.
I suggested a revision according to the Instruction for Authors as concerns the references in the text. In addition, I suggest a revision for typing errors
Author Response
Thank you so much for your valuable feedback. Please see our detailed responses to your comments here.

Reviewer 2 Report
Comments and Suggestions for Authors
Article title: Common Beverage Consumption and Benign Gynecological Conditions
The topic of the review article is highly actual, it could be interesting also for non-experts in the field. The article is written in a comprehensible way, with a sufficient amount of literature, including the latest information from the described field.
Comments:
- Line 216: caffeine and caffeic acid are completely different compounds. If we are talking about caffeinated beverages, the definition is: Caffeinated beverages are drinks containing caffeine. Caffeic acid is a hydroxycinnamic acid, with free hydroxyl groups, and i tis know as antioxidant with many beneficial properties on the human body. I recmmend to exclude information about caffeic acid, and focu only on caffeinated beverages.
- I recommend to delete also figure 2 – it can be confusing for readers. And replace it with the effect of caffeine (not caffeic acid).
- Line 293: information about resveratrol. This part could be also excluded from the article, it is not topic of this review. Resveratrol is secondary plant metabolite, polyphenol, antioxidant, with bebeficial effect on human body. It is well known that it is present in red wine – see also French paradox. But if the authors discuss the alcohol influence, it cannot be associated with the effect of this polyphenol.
- Line 306: Informations about data on effect of selected bevergas on PCOS.
In this section, the informations describes the relationship of drinking beverages in relation to fertility. However, fertility can be affected by many factors, not only PCOS. Therefore, I recommend rewriting this part of the article, especially caffeinated beverages.
- Conclusion: Table 1. According informations reviewd in the manuscript, I can not agree with this table. Because many informations are not clear, are not fully understood, or more research is needed, or there is lack of clinical data...I suggest to redo table, and include all informations mentioned in manuscript (not only unfavorable, no strong association, or beneficial association).
- Some typo mistakes in article.
- Please double check literature formatting.
Author Response
Thank you so much for your valuable feedback. Please see the attached document with detailed responses to your comments.

Reviewer 3 Report
Comments and Suggestions for Authors
This paper gives an review of common beverage consumption in relation with gynecological conditions with the goal to increase the transparency of the possible effects of these four beverages on five benign gynecological conditions.
The work has a clear structure, the only thing I suggest is to explain again the abbreviations of benign gynecological conditions (UF, PCOS & PD) under table 1.
Congratulations to the authors for the idea and the writing.
Author Response
Thank you so much for your valuable feedback. Please see our detailed responses attached.
